# Traditional Microscopic Techniques Employed in Dental Adhesion Research—Applications and Protocols of Specimen Preparation

**DOI:** 10.3390/bios11110408

**Published:** 2021-10-21

**Authors:** Agnieszka Nawrocka, Ireneusz Piwonski, Salvatore Sauro, Annalisa Porcelli, Louis Hardan, Monika Lukomska-Szymanska

**Affiliations:** 1Department of General Dentistry, Medical University of Lodz, 251 Pomorska Str., 92-213 Lodz, Poland; agnieszka.nawrocka@stud.umed.lodz.pl; 2Department of Materials Technology and Chemistry, Faculty of Chemistry, University of Lodz, 163 Pomorska Str., 90-236 Lodz, Poland; ireneusz.piwonski@chemia.uni.lodz.pl; 3Dental Biomaterials, Preventive and Minimally Invasive Dentistry Departamento de Odontología, Facultad de Ciencias de la Salud, Universidad CEU-Cardenal Herrera C/Del Pozo ss/n, Alfara del Patriarca, 46115 Valencia, Spain; salvatore.sauro@uchceu.es; 4Department of Therapeutic Dentistry, I.M. Sechenov First Moscow State Medical University, 119146 Moscow, Russia; 5Department of Dentistry, Catholic University of the Sacred Heart, 00168 Rome, Italy; annalisaporcelli97@libero.it; 6Department of Restorative Dentistry, School of Dentistry, Saint-Joseph University, Beirut 1107 2180, Lebanon; louis.hardan@usj.edu.lb

**Keywords:** microscopy, CLSM, SEM, TEM, AFM, dental adhesion

## Abstract

Microscopy is a traditional method to perform ex vivo/in vitro dental research. Contemporary microscopic techniques offer the opportunity to observe dental tissues and materials up to nanoscale level. The aim of this paper was to perform a literature review on four microscopic methods, which are widely employed in dental studies concerning the evaluation of resin-dental adhesive interfaces—confocal laser scanning microscopy (CLSM), scanning electron microscopy (SEM), transmission electron microscopy (TEM) and atomic force microscopy (AFM). The literature search was performed using digital databases: PubMed, Web of Science and Scopus. On the basis of key words relevant to the topic and established eligibility criteria, finally 84 papers were included in the review. Presented microscopic techniques differ in their principle of operation and require specific protocols for specimen preparation. With regard to adhesion studies, microscopy assists in the description of several elements involved in adhesive bonding, as well as in the assessment of the condition of enamel surface and the most appropriate etching procedures. There are several factors determining the quality of the interaction between the substrates which could be recognized and a potential for further implementation of microscopic techniques in dental research could be recognized, especially when these techniques are used simultaneously or combined with spectroscopic methods. Through such microscopy techniques it is possible to provide clinically relevant conclusions and recommendations, which can be easily introduced for enamel-safe bonding and bonding protocols, as well as optimal pretreatments in dentine preparation.

## 1. Introduction

Although the morphology and modification of dental tissues have been a matter of interest for investigators throughout the centuries, the real onset of dental research began in the mid-1940s [1]. The number of dental specialists has increased over the years and the prevalence of dental diseases has become more visible. Thus, the Public Health Service in the USA highlighted the need for financial support in dental research [1]. In the 1950s, studies gained an intensity even including pioneering microscopic research with modern and crucial technology [1].

Microscopy still represents one of the most fundamental techniques in dental laboratory research. However, the present technological development goes far beyond the combination of visible light and the system of lenses. Limitations of the conventional optical imaging were overcome with the invention of electron and scanning probe microscopy [1,2,3]. Contemporary microscopes use various sources of energy to obtain images and offer incomparably greater magnification and image resolution (Table 1). This gives the opportunity to assess the interface between dental tissues and materials up to nanoscale level.

Researchers need to answer numerous questions to choose the optimal microscopic techniques in their studies. The first factor to consider is the type and the composition of specimens. Solid, inorganic material requires different preparation compared to soft, organic tissue prone to damage during observation. The research design must determine if the particular specimen would be reused in the sequence of observations. Some pre-observation protocols are irreversible, that is, the material cannot be investigated in a different context again. Secondly, it should be defined which details are crucial to fulfill the aim of the study: general overview of specimen morphology, surface characteristics, internal structure or micromechanical properties. Properly chosen magnification and resolution reveals the necessary details. Imaging of intersurface relationships may require contrast enhancement by means of specific staining methods; also, not without significance is the size the of the specimen, especially its thickness. Taking these aspects into account, there is a need to summarize the principles and possible implementations of microscopic techniques [1,2,3].

The aim of this paper was to perform a literature review on the current microscopic methods mainly employed in dental research. In the context of dental adhesion, particular attention was paid to four techniques: confocal laser scanning microscopy (CLSM), scanning electron microscopy (SEM), transmission electron microscopy (TEM) and atomic force microscopy (AFM) that can precisely depict each element involved in dental adhesion. The comparison of microscopic methods and manifold implementations are presented in the Table 1. In particular, this paper highlighted the physical background of methods mentioned and the protocols of proper preparation of dental specimens so as to achieve optimal microscopic images.

## 2. Search Strategy

The literature search was performed using the PubMed, Web of Science and Scopus digital databases using the keywords relevant to the topic: “microscopy”, “CLSM”, “SEM”, “TEM” and “AFM”, as well as the full names of the mentioned abbreviations. These terms were combined with the phrases: “dental”, “dental adhesion”, “dental research” and “dental materials” using the basic Boolean operators: “AND” and “OR”, as follows: (CLSM) OR (confocal laser scanning microscopy) AND (dental) OR (dental adhesion) OR (dental research) AND (SEM) OR (scanning electron microscopy) AND (dental) OR (dental adhesion) OR (dental research) AND (TEM) OR (transmission electron microscopy) AND (dental) OR (dental adhesion) OR (dental research) AND (AFM) OR (atomic force microscopy) AND (dental) OR (dental adhesion) OR (dental research). The initial results included 10,861 articles, which, after removal of duplicates and analysis using inclusion and exclusion criteria, were finally reduced to 84 articles (Table 2).

## 3. Confocal Laser Scanning Microscopy (CLSM)

### 3.1. Historical Background

The concept of confocal microscopy provided a major step forward in fluorescence imaging. The primary idea was presented in the 1950s by Minsky and was developed ten years later by Egger and Petran. The aim was to reduce out-of-focus signal distortion to improve image quality. In 1987, with the advancement in computer and laser technology, the first commercial confocal laser scanning was presented and it became widely implemented, especially in microbiological research [4].

### 3.2. Principles of Technique

Confocal laser scanning microscopy (CLSM) is a type of fluorescence microscopy that provides a significantly better image resolution and contrast compared to the conventional light microscope. Light waves that are the source of energy for specimen excitation are passed through a small aperture located on the confocal plane (Figure 1). The beam of light is more focused (pointed illumination) so the diffused out-of-focus signal is eliminated. The specimen absorbs the energy and responds by emitting longer light waves. That emission is registered by the detector and converted into an image. The advantage of CLSM is also attributed to the possibility of performing optical sectioning (creating images of thin slices of a specimen by eliminating out-of-focus light in each plane) to generate three-dimensional reconstruction of specimens [4,8].

### 3.3. Specimen Preparation

To observe dental tissues in CLSM, thin sections of specimens need to be prepared [9]. The sectioning procedures create the smear layer, which consists of denatured cutting debris accumulated on the specimen surface. Therefore, pretreatment is required to remove the debris and obtain a clear image. In some specific situations, the pre-observation protocol includes the combined usage of sodium hypochlorite solution, ultrasonication and vacuum. Optimal results may also be achieved when the specimens are stored for 1 h in 5.25% NaOCl solution activated by ultrasounds. This provides successful visualization reaching a depth of 38 µm [9]. A representative pretreatment protocol is presented in Table 3 [8,10,11].

In the CLSM technique, imaging is based on fluorescence, so the specimens need to be labelled with a specific substance containing a fluorophore. The most common dye is rhodamine because its absorption peak (511 nm) is compatible with the excitation wavelength used in most confocal laser microscopes [9,12,13]. The best effectiveness among different staining protocols provides application of the rhodamine 123 in concentration 10^−5^ M and rhodamine B (10^−5^ M) for 3 h [9,10,11]. The specimen then needs to be rinsed 2–3 times and stored in demineralized water for 5 min. Finally, the specimen can be fixed to the glass slide with cyanoacrylate glue and observed. The other possibility is to use a dye to observe its interaction with dental tissues [8,12]; for instance, by mixing 0.1 wt.% rhodamine B with tested etchants [8].

### 3.4. CLSM in Dental Adhesion Research

From the first implementation performed in 1987 [4,11,12], CLSM became a suitable method for dental research. According to the PubMed MEDLINE almost 390 papers containing words “CLSM” and “dental” were published during the last decade and after exclusion of those publications not relevant to the topic of dental materials (for instance—microbiological studies)—200 articles were found. The aim of precursory CLSM studies conducted over 30 years ago, was to evaluate the extent of the dentin smear layer at the microscopic level, as well as the structural organization of dentin tubules. Several further studies used CLSM to visualize the hybrid layer and restoration-dentin interface [4,11,12,14,15] or to assess the properties of new generation composite materials [13]. Their main objectives were to establish the optimal protocols of bonding techniques and to improve the quality of adhesive connection. Research also proved that the thickness of hybrid layer was correlated with etching time [12]. Among various fluorescent dyes, rhodamine is considered as the most adequate in adhesion studies, because it contains electrically neutral, undissociated molecules, that are insoluble in water but extremely soluble in organic solvents. The components of adhesive systems (primers, sealers and silanes) create the environment for an equal distribution of a dye [13]. Rhodamine B is water soluble and remains stable irrespective of pH conditions. It exhibits a red color, maximal excitation at a wavelength of 556 nm and a maximal emission at 580 nm [13]. These properties enable the observation of the interface area between the adhesive material and dental tissue and reveal how deeply the primer incorporates intertubular dentin and penetrates the dentin tubules [4,11,12]. Further dyes, such as fluorescein isothiocyanate (FITC) or acriflavin [12,14], were also implemented in research. FITC has a green color and its excitation and emission peak is set at wavelengths 495 nm and 519 nm. The values characteristic for acriflavin are, respectively, 436 nm and 520 nm. However these solutions do not perfectly identify the components of the hybrid layer due to their premature leaching [12]. Dávila-Sánchez et al. [14] presented the CLSM images of an acid-etched dentin surface which was additionally exposed to the experimental flavonoid solutions (Figure 2). The aim was to assess the influence of applied pretreatment protocol on the interaction between dentin and bonding system. The specimen preparation was as follows: adhesive system mixed with the water-soluble rhodamine B and applied on the dentin surface (pretreated with various flavonoids). After polymerization, the cross-sections of tooth specimens were made and subsequently immersed into a fluoresceine solution. The infiltration of fluoresceine visible in the CLSM image served as an indicator of nanoleakage in the hybrid layer and dentin tubules [14].

Understanding the effective methods of dentin remineralization is also the aim of numerous research. A novel approach is based on the of bioactive nanofibers applied in dentin to reduce hypertensives (DH). CLSM performed in fluorescence mode enabled the quantitative analysis of intertubular obliteration (Figure 3) and proved that the application of nanofiber scaffold was a promising treatment for DH [16].

CLSM studies (combined with SEM) were also used to observe the infiltration zone of adhesive within the etched enamel [13,17]. Because the acid etch technique is considered as the gold standard in dental bonding, the visualization of how the 37% phosphoric acid (H_3_PO_4_) interacts with enamel surface is important. The extent of tissue loss varies in different research with an average value of 3 µm [17]. Even slight dissolution can coexist with tissue alterations reaching the depth of 100–200 µm. The actual range of tissue “alteration” differs depending on the quality of enamel and its initial mineral content [17]. After exposure to acid, enamel gains a structure favorable for microretention and subsequently the applied adhesive can penetrate surface irregularities [13,17]. Microscopy gives the possibility of observing the action and invasiveness of different etching protocols, such as the self-etching technique (SET). Researchers, searching for an alternative for aggressive acid etching, used CLSM to observe how acidic monomers in the self-etching primers interact with dental hard tissue [17]. CLSM revealed different post-etching enamel patterns compared to the acid etch technique (AET). The resolution and the magnification provided by the microscope enabled visualization of even a slight demineralization—reaching only a few micrometers into the superficial enamel layer (Figure 4). The use of fluorescent dye may led to obtaining precise images to show how primer molecules could penetrate the structure with concurrent etching action [8,17].

An approach to find an efficient etching agent without side effects encouraged investigation of novel bonding products described as “enamel safe” [9]. The action of a new etchant was to dissolve the superficial layer of enamel along with inducing CaP reprecipitation in enamel prisms. Remineralization was achieved with β-tricalcium phosphate (β-TCP) and monocalcium phosphate monohydrate powders (MCPM) added to conventional etchants: citric acid or 37% phosphoric acid. The additional components served as the precursors of CaP and prevented potential enamel damage. CLSM with oil-immersion lens and a laser wavelength of 568 nm gave the images of 63× magnification. That revealed undefined post-etching patterns with dispersed micropores partially filled with re-precipitated film. Results were supported by observations from other microscopic and spectroscopic techniques (SEM, XRD, Raman spectroscopy) and led to the conclusion that the novel etching protocol with CaP reprecipitation provides an enamel surface less susceptible to bonding and debonding procedures [9]. However, the benefit of using the CLSM technique was not focused on the “simple” information about interaction between enamel and etchant. Indeed, the image also visualized the remineralization abilities. The higher degree of fluorescence was correlated with increased demineralization [9]. Rhodamine excited by a particular laser wavelength was detected and its location depended on the enamel composition. More visible dye color indicated that more enamel prisms were exposed with a remarkable loss of organic component. Conventional etching agents based on 37% phosphoric acid created a honey-comb enamel pattern and its fluorescence was significantly higher in comparison with “enamel safe” etchants [9].

Valuable results were also achieved in research combining confocal scanning microscopy with optical coherence tomography [18]. The hybrid of two non-invasive methods enabled generation of high-quality 3-D images of enamel-bracket interfaces in orthodontic bonding. Any material defects, disturbances in adhesive resin, gaps and interphase imperfections were revealed. The presence of the disturbances is an important predictor for quality of orthodontic bonding. The implementation of this research into the clinical practice requires further investigation, although the precision of imaging is promising. It can be used to determine the maximal acceptable value of gaps in the material–enamel interface and can become a foundation for further dental materials research [18].

### 3.5. Limitations

Compared to SEM, TEM and AFM, the maximal magnification achieved by CLSM is significantly lower. Taking into consideration the mentioned microscopic techniques, CLSM can be considered as a method able to provide only the general overview of the studied surfaces, without the possibility to capture the details in nanoscale [4,8]. Despite the fact, CLSM can achieve high spatiality and image quality can be lower in thick specimens. Increased specimen size and internal heterogeneity can cause light beam dispersion from the focal plane to the objective lens. The spherical aberration occurs because of the lack of congruence in the refractive index. The point-spread function broadens and the value of signal peak is lower. In other words, the deeper the imaging, the greater the image distortion in the z-axis. To achieve the best image quality, the depth of the specimen should not exceed 100 µm [19]. Further issues can also be related to the disposition of fluorophores; as a result, fluorescence emission signals can overlap and cause inaccuracies [20].

### 3.6. Future Perspectives

Materials is dentistry are constantly modified and improved to obtain the best combination of biological, chemical and mechanical rheological properties and to improve adhesion. CLSM with non-invasive and undemanding specimen preparation can provide precise imaging of material and with no risk of changing its properties. An important approach in material science is surface modification and CLSM was successfully implemented in such non-dental research. Indeed, Li et al. [21] used this technique to observe the modified surface of nanochannels. Fluorescein connected with the superficial amino-groups, resulted in visible fluorescence effect and proved that surface modification was successful [21]. Such methods for surface modification have the potential to trigger novel trends in dentistry. Because CLSM can provide three-dimensional reconstruction of the observed material, it is possible to visualize the spatial structure of polymers. The dispersion of inorganic fillers in organic matrix within resin-based composites plays a pivotal role in restorative materials. Observations preformed with electron microscopy have some limitations in that area, providing only 2-dimensional images with a limited depth of observation and requiring particular specimen preparation. CLSM is an optimal alternative and can be implemented in two possible ways. The first is called pre-labelling, when the filler is marked with fluorophore before incorporating the organic matrix. Researchers are developing methods of pre-labeling that use a specific surfactant with an aggregation-induced emission-active fluorophore. The second method, described as post-labelling is dedicated to observing assembled material, when the composite is stored in fluorescent dye [21]. The future perspectives of CLSM in dentistry depend on the introduction of new fluorescent dyes and improvement of optical elements.

## 4. Scanning Electron Microscopy (SEM)

### 4.1. Historical Background

The SEM technique derives from a group of electron microscopy and is the result of the development of the pioneering transmission electron microscope. Because of the common origin of these methods, their historical background is usually described together. The invention of electron microscopy is attributed to Max Knoll and Ernst Ruska, the scientists from Technical University of Berlin, who in 1931, obtained a series of images from an electron beam scanner. The first high-resolution transmission electron microscope was presented six years later by von Adrenne and in 1938 it became available for commercial use. Simultaneously, at the beginning of the 1940s, Zworykin et al. performed the first SEM observation of solid specimens [2]. Only a decade later the development of the technique along with the progress in specimen preparation enabled producing 3-dimensional images. Throughout the years the sources of energy became more effective as well as the detectors to capture the secondary electrons. The 21st century innovations included the digital processing of images with the aid of computer technology and the attempts to perform the observation in more natural conditions without the requirement of vacuum environment and the specimen fixation [2,16,17,21].

### 4.2. Principle of the Technique

Unlike conventional optical microscopy, SEM does not use visible light to illuminate the specimen. As a source of energy, a beam of electrons is generated (Figure 5). Since the length of electron waves is 10^5^ times shorter than visible light waves, it is a technique that gives the opportunity to obtain images with significantly higher resolution. Indeed, the observer can achieve 10^7^× magnification, whereas in light microscopes that value is limited and the maximal achievable magnification is only 2000× [22].

The beam of electrons comes into contact with the specimen molecules and the result of that interaction is an energy release in different mechanisms such as: back-scattered electrons (BSE), secondary electrons (SE) (low-energy or high energy), cathodoluminescence or X-ray emission [22].

### 4.3. Specimen Preparation

Since dental tissues are nonconductive specimens, they require special preparation so as to be observed in SEM without electrostatic charge. When the specimen is exposed to the electron stream, the artifacts in the image occur quite commonly. To avoid such an issue, the specimen needs to be covered with a film of electroconducting material. An extremely thin coating layer is usually composed of gold, but also other materials are used in research, such as noble metals (palladium, platinum, iridium or osmium), wolfram or graphite. The covering is spread on the surface by means of sputter deposition [22,23].

A further requirement is the specimen dryness. In the vacuum environment, which is used in the conventional SEM imaging, the presence of water may generate disturbing surface tension leading to alterations in surface morphology [24]. Consequently, soft-tissue specimens (with significant water content in their internal structure) need to be fixed with chemical agent (mostly glutaraldehyde) and then dehydrated. The need for fixation of dental specimens is questioned. Mineralized hard tissue like enamel is relatively dry due to its very low water content and it does not require special fixation procedures. Although dentin contains more organic substance than enamel and its collagen network appears to be more susceptible to collapse during dehydration, according to research, these structures can withstand air-drying undestroyed, even without fixation [24]. The dry and coated tooth specimen is finally placed on the stub with electrically conductive carbon tape.

Despite the electron beam being able to interact with the specimen in different mechanism (as was it mentioned in paragraphs above), the most valuable signal used in dental research is secondary electrons (SE) [24]. These electrons are released from the atoms of the specimen surface and their energy is transferred into a readable image. The high resolution of an image is achievable due to the small diameter of the initial election beam. Due to accurate mapping of morphological features of surfaces, the imaging was used in research to analyze the biofilm on enamel, as well as the condition of hard tissues after abrasion, polishing, etching or bonding [6,23,24]. Another source of energy used in SEM imaging in dental studies are back scattered electrons (BSE). The process of visual representation is based on the electrons that were reflected from the specimen surface and captured by the detector. The number of back-scattered electrons is correlated with the atomic composition of the tested specimen, thus it is a useful tool to evaluate the chemical distribution, especially in a dentin–resin hybrid layer [23]. However, the image resolution compared to SE is lower. BSE in dental research was implemented not only in qualitive observation, but also in quantitate evaluation determining mineral density in dental tissues [23].

An SEM image without alterations depends on a further significant parameter—the input energy. To release an electron beam from the source electrodes, the proper voltage must be applied. Higher voltage means a higher electron beam energy and, as a result, a higher propagation within the specimen [16]. Some authors recommend 15–25 kV acceleration voltage as a value providing a suitable resolution and brightness differentiation in dental research [24]. For instance, it is possible to distinguish a composite restoration from adhesive resin at lower magnifications. However, high voltage values could also have adverse impacts on tested specimens. During the interaction, the temperature increases and may cause the evaporation or sublimation of organic components [23]. Low-voltage SEM gives the possibility of observing nonconducting specimens without coating. The balance between incoming and outcoming electrons is important. Despite the advantages, low voltage acceleration is not routinely implemented in dental research. Teeth, as biological specimens, are composed of elements with low atomic numbers. As a result, low voltage generates the secondary beam with insufficient intensity that causes signal aberrations [24].

The modification of SEM described as environmental scanning microscopy (ESEM) is an attempt to overcome the limitations in biomedical usage. It allows observing of undried specimens and without prior coating [25]. The specimen is placed in a chamber in a strictly controlled gaseous atmosphere. The source of energy—primary electron beam—is similar to conventional SEM and it interacts with the specimen surface with a release of secondary electrons. However, in ESEM an interaction between gas molecules and electrons also occurs, which is called ionization. This process leads to additional enhancement of signal which is transferred to the detector. Because of the presence of the positive gas ions, the undesired specimen charging can be avoided [25].

### 4.4. SEM in Dental Adhesion Research

Invented in 1931, scanning electron microscopy quickly became a fundamental imaging technique in biomedical studies [2,25,26]; an example of its feasibility in dental tissues imaging is presented in Figure 6 and Figure 7 [27,28].

The search in PubMed/MEDLINE database revealed over 3000 articles when including terms such as “scanning electron microscopy” and “dental” only from the last decade. The huge peak in the number of publications was registered in 2010. The main applications of SEM in 2020 in dental studies can be divided into the categories presented in Table 4.

In adhesion research, a special consideration is given to the influence of orthodontic bracket debonding on enamel surfaces [17,29,30]. Debonding in orthodontics, considered as removal of devices from dental tissues, must be performed in as harmless a manner for enamel as it is possible. Depending on the quality of enamel-bracket connection, different types of bonding failure can occur (adhesive, cohesive or structural). It can be observed clinically as an amount of residual adhesive on the enamel surface and assessed using the adhesive remnant index (ARI). SEM enables assessing of the enamel surface after debonding and defined by ARI score with great accuracy. Numerous microscopic researches were focused on finding potential enamel cracks, scratches and possible undesirable exposure of enamel prism endings [29]. These observations are possible in the magnification and resolution provided by SEM. The unflawed superficial enamel layer is more resistant to caries; thus, its iatrogenic damage may cause post-treatment complications. Several observations may help to compare ARI scores after various bonding/debonding protocols, as well as to find the safest methods for enamel for resin/cement remnant removal. Combining SEM with 3D profilometry, researchers obtained the full enamel surface profile and 3-dimensional measurements of remnants and materials debrides [30]. Studies revealed the enamel loss was caused by all prebonding protocols (pumicing, etching) and debonding procedures (following material removal). However, the cumulative loss is relatively low in comparison to the total enamel thickness [17]. The incorporation of adhesive into the etched surface to the depth of 20 µm is undesirable and potentially harmful. After removal of brackets and cleaning procedures, the residual material remains within the tissue, thus causing a further alteration of enamel prisms [17]. This aspect, confirmed by SEM observations, gives the advantage to self-etching systems, because after their application, the risk of enamel incorporation is lower [17].

An important research area is the evaluation of nanoleakage at the dentin–resin interface. Nanoleakage affects long-term adhesion and it can be revealed in SEM using the silver nitrate penetration method [31]. Before resin application, tooth specimens are stored in a silver nitrate solution, since it contains ions serving as a tracer to improve the contrast in microscopic images [31,32,33]. Silver nitrate molecules, because of their small dimension, can effectively diffuse into the gaps within the hybrid layer. According to research, the most frequent protocol is specimen storage in total darkness for 24 h in a solution of 50% *w*/*v* AgNO3 (pH = 6). Specimens are then rinsed with water and exposed to fluorescent light for 8 h to convert silver ions to metallic silver.

To improve the analysis of the specimen’s topography with information of its elemental composition, it is convenient to combine microscopic and spectroscopic methods. SEM can be combined with X-ray microanalysis called energy dispersive spectrometry (EDS) [22,24,34]. The electron beam generated by the microscope interacts with the specimen. As a result, the effect is not only a release of secondary electrons and back-scattered electrons (used for specimen imaging), but also an induction of X radiation. The X-ray beam is registered by the spectrometer. The obtained spectrum depends on the specimen composition and change in the mineral component is captured [22,24]. The method was used in testing various pre-restoration protocols and its influence on shear bond strength (SBS) [28,35,36]. The change in the elemental composition in dentin–resin interface confirmed that the implemented protocols had an impact on SBS [27,35,36].

### 4.5. Limitations

The main disadvantage of SEM is related to the destructive specimen preparation. Because of the vacuum conditions, specimens require primary fixation and dehydration of organic specimens. Moreover, non-conducting specimens need to be sputter-coated to avoid electron charging. The images obtained with conventional SEM are 2-dimensional and measuring the depth of an object is a challenge [17,29]. Obtaining the third dimension is possible after digital reconstruction, which is described in further paragraphs.

## 5. Transmission Electron Microscopy (TEM)

### 5.1. Historical Background

The TEM technique is described as a predecessor of SEM, although through the decades both techniques evolved, thus finding their specific implementations in dental research. With the development of high-resolution objective lenses in the 1970s, imaging techniques evolved. Particularly important was the improvement of the field emission electron gun, that provided increased spatial resolution and better noise-to-signal ratio [2,16,17,21,37,38,39].

### 5.2. Principle of Technique

Similar to SEM, transmission electron microscopy also uses a beam of electrons as a source of energy. However, the mechanism to obtain an image is different in TEM compared to SEM. In TEM, the electrons are transmitted through the specimen and then registered. Electrons are passing and interact with the specimen generating secondary elements—diffracted and transmitted electrons that are captured by a detector [39]. The electron beam used as a source of energy requires vacuum so as to avoid the electrical discharge of the electron gun and to ease the beam transmission. TEM mainly consists of an electron beam generator, a configuration of different types of lenses and detector (Figure 8) [40].

The unique feature of the TEM technique is the opportunity to obtain not only the topographical characteristics of the specimen but also the visualization of its inner structure [40,41]. That is why TEM is nowadays applied in a broad field of biomedical research; for instance, the observation of molecular structures of cells and tissues and for description of numerous processes at cellular level [39].

### 5.3. Specimen Preparation

TEM observation requires a vacuum environment for observations and a specific and sophisticated protocol for specimen sectioning and fixation is required [41]. The first important aspect is the size of the specimen. In thicker specimens, the fixation agent cannot infiltrate the structure effectively. Thus, ultrathin sections created with the ultramicrotome are recommended to enable an effective passage of electrons thorough the specimen [42]. Suggested size of the tissue block should not exceed 1 mm^3^. In large specimens, over and under-fixed areas can be found difficult to image. Moreover, a suitable protocol for fixation avoids alteration of the mechanical properties of the specimens. The effectiveness of fixation depends on the components of the fixative agent [42]. According to Park et. al., when the 2.5% glutaraldehyde is used the fast fixation range reaches 0.13 mm and the maximum tissue size should not be larger than 0.25 mm. When the mixture of glutaraldehyde and formaldehyde is applied—the area of the fast fixation reaches 0.5 mm and the maximum specimen thickness should not exceed 1 mm [41,42]. As it was mentioned in the SEM technique, the proper tissue fixation prevents degeneration of the material during its dehydration. The most common method of fixation are chemical agents. There are the two most common methods of fixation. First (recommended for dental hard tissues) is immersion, when the specimen is just stored in the chemical solution in particular environmental conditions. The second method, applied mostly in laboratory animals, is based on the perfusion of chemical agent through the blood vessels and cannot be implemented in dental material science [42]. The main fixative agent used in dental research is 2.5% glutaraldehyde solution in 0.1 M phosphate buffer. The effective fixation protocol was described by Arana-Chavez et al. [43]. The tooth is stored in the buffered solution containing 2.0% glutaraldehyde and 2.5% formaldehyde. The time of storage depends on the density and the size of the specimen and the optimal time for dental hard tissues is 6 h at room temperature. Alternatively, to increase the diffusion of fixative agent into the specimen, the immersion can be supported with microwave irradiation [43]. Optimal visualization of macromolecules, such as dentinal collagen, can be achieved via negative staining method. The procedure was developed in the 1960s, in the same period as the introduction of TEM in dental research. Firstly, the specimen is attached to the support substrate (usually amorphous carbon). Further steps include an application of the electron dense stain, blotting and making the surface dry. The stain accumulates on the surface creating a thin film. The specimen has lower electron density in comparison to the staining agent and appears as the lighter area. The method is easily applicable, but its limitation is the resolution restricted to 18 Å. A superficial layer is created due to the repulsion of similarly charged molecules of specimen and staining agent. To enhance the visibility of collagen, the most frequent staining agent is methanolic uranyl acetate [44].

### 5.4. TEM in Dental Adhesion Research

TEM was used for accurate visualization of the internal structures of the specimen. For instance, it is possible to observe the nanocrystals of hydroxyapatite in human enamel [45], dentinal constituents [46] and the differences in mineral content at the dentino–enamel junction [39]. The images of adhesive–dentin ultrastructure were obtained in the research evaluating the effectiveness various bonding protocols [47,48] (Figure 9). The methods improving dentinal remineralization were also presented [49]. In PubMed, over 1000 papers can be found concerning the subject of TEM application in various fields of dentistry that were published in years 2011–2021.

Numerous contemporary studies in material science are focused on the possible clinical implementation of various remineralization agents. Observations conducted with TEM confirmed that application of synthetic hydroxyapatite nanoparticles is regarded to improve the enamel resistance caries [50,51]. Synthetic HAP were also added to the adhesive solutions to achieve higher shear bond strength [52]. In another study, TEM supported with CLSM was used to observe the remineralization effect of ion-releasing materials. Although the results showed that caries lesion in dentin cannot be totally remineralized, images proved an important mineral precipitation [39].

Alternative etchants implemented in recent years were also analyzed via TEM [53,54]. TEM microphotographs provided the overview of the differences in hydroxyapatite dissolution after application of various etchants. Combining TEM findings with SEM observation and µTBS test, the researchers reached the conclusion that etchant based on ZrO(NO_3_)_2_ can serve as an effective alternative to conventional etchants based on phosphoric acids [53]. Several observations of ultrastructure of various restorative materials were also performed with TEM [55].

In the context of dental adhesion, nanoleakage within the hybrid layer was observed in numerous TEM studies conducted by several researchers [15,56,57]. Their researches were based on the assumption, that the dentin–resin interface contains microporous zones that can be revealed by their infiltration with silver nitrate tracer (sliver nitrate penetration protocol was described above, according to SEM observations) [56]. Various adhesive systems were tested to reveal nanoleakage. Researchers presented the concept of water trees as a possible cause of the long-term adhesive degradation. The “water trees” observed in TEM may represent the propagation of residual water within the interface and that released from hydrophilic monomers used in self-etch systems [15,56,57].

A negative staining technique was implemented in studies focused on the properties of dentin collagen. Clinically relevant studies are concerned with biomineralization—the process of induced dehydration of the collagen matrix and replacement of the internal water with apatite crystallites. Kim et al. performed the observation of human dentin, which was conventionally etched and filled with adhesive resin. A sliver tracer was used to reveal the residual water within the hybrid layer in TEM observation. Subsequently, the specimens were stored in a biomimetic remineralization medium. After a year of storage, randomly selected specimens were stained with 2% methanolic uranyl acetate and aqueous lead citrate for TEM observation. In comparison to the control group of specimens stored only in the simulated body fluid (SBF), the images of specimens that were kept in biomimetic agent revealed extensive areas of intrafibrillar collagen remineralization. The positive effect was also confirmed with microtensile strength tests—the specimens with apatite-incorporated collagen revealed significantly higher bond strength results. The development of a clinically applicable biomimetic agent may significantly improve the longevity of restorations [58].

### 5.5. Limitations

The disadvantages of TEM in the context of invasive preparation of non-conducting specimens are similar to SEM and are widely described in previous paragraphs. Another limitation in the observation of dental tissues is the need of obtain ultra-thin sections (80–90 nm) that require in-depth operator expertise. The thickness of the specimen determines the resolution and the quality of the image [58,59].

Moreover, the visualization of complex organic molecules requires contrast enhancement and it is connected with additional specimen staining (e.g., negative staining technique) that increases the time and difficulty of preparation as well as the cost for the overall process.

### 5.6. Future Perspectives of SEM and TEM

The enhancement of image resolution is only one aspect in the development of electron microscopy. Imaging technology is constantly being improved by using more effective sources of electron beam, accelerating voltage and aberration correctors [60]. However, the limitation of conventional electron microscopy in dental adhesion research is the two-dimensional image, while 3D visualization of the inter-surface area is needed for better understanding of the nature of adhesive connection. Although the first attempts to reconstruct the 3D geometry of flat objects were taken in the 1960s, only the digital development in the 21st century provided sufficient precision to implement spatial reconstruction into electron microscope observations [60]. Modern methods include the multi-view algorithm that is based on multiple images from different perspectives [61]. In electron tomography (ET), the specimen spins around a fixed axis while the microscope chamber tilts with the maximal inclination of 70 degrees. Nowadays, the great challenge is to achieve atomic-resolution of 3D reconstruction, especially in biological specimens. Contrary to inorganic specimens, the images of organic items do not always provide satisfactory contrast, since they are often prone to being damaged by the electron beam [62,63]. Along with future developments in 3D imaging and great contract quality, the current methods for biological specimen preparation and preservation are required.

## 6. Atomic Force Microscopy (AFM)

### 6.1. Historical Background

Atomic force microscopy (AFM), invented by Binning and Rohrer in 1986 [3,7,64], originated by a method called scanning tunneling microscopy. The procedure was based on the use of a scanning probe to map a surface at atomic level. The first nanoprobe was designed in 1991 and the first commercial AFM was available in 1998 [3,7,64].

### 6.2. Principle of the Technique

AFM enables the acquisition of very high-resolution image and its main advantage is the possibility to visualize nonconducting specimens under normal pressure conditions and without additional surface coating. Such a feature is useful in biomedical research, i.e., dental studies [3,7]. Observations can be performed in liquid environment without the risk of dehydration; that decreases the risk of potential artefacts correlated to fixation and drying procedures [3]. An example of images of enamel obtained via AFM is presented in Figure 10 and Figure 11 [28,65]. The applicability of AFM also includes nanomechanical measurements such as determination of Young’s modulus (Figure 11B) and mean adhesion forces between scanning probe and specimen surface (Figure 11C).

The microscope is equipped with a cantilever and a tip (a probe) to scan the specimen surface (Figure 12). The main principle is the interaction between the tip and the atoms present in the tested specimens. The source of the effect are intermolecular forces (for example van der Waals, chemical bonds, electrostatic forces) that generate the deflection of the tip. The changes in the tip deflection are processed via piezoelectric scanner and the detector (photodiode) into an electric signal. Basically, the laser beam is focused on the back part of the tip so the light is reflected and directed to the photodiode [66]. Changes in tip-specimen conformation leads to the different position of the light spot on the photodiode, which is a signal source. The signal is transferred to the corresponding device and a topographic image is generated [3,7,66]. The surface of dental tissues and materials can be observed at nanometric scale to obtain information about topography and micromechanical properties.

AFM can work in different modes: contact mode, tapping mode or non-contact mode. The first two are the most frequently used in dental research. In contact mode, the probe remains in touch with the solid surface throughout scanning process. It is used to receive a precise topography of the tooth surface. However, the constant contact between tip and specimen can cause significant forces resulting in image artifacts, especially when a delicate surface is observed. Tapping mode is usually dedicated to air or liquid conditions and the tip oscillates, making periodic contact with the surface [7]. Because the interaction between the tip and specimen is only intermittent, the potentially damaging intermolecular forces are mitigated and this mode can be used to observe fragile surfaces (i.e., the etched enamel coated with an adhesive bonding agent) [7].

Force volume mapping (FVM) gives the possibility of characterizing nanomechanical properties of specimens, including adhesive forces, viscoelasticity and Young’s modulus as well as chemical forces and dielectric conductance. The probe works as an indenter inducing vertical forces. In order to generate such measurements, the feedback loop should not be connected when the probe is moving along the specimen’s surface; the changes in vertical deflection of the tip are registered [67]. Mapping is performed by gathering multiple force curves over particular regions of the specimen within a defined network. Force–distance curves represent (graphically) the force applied on the specimen in correlation with the tip–specimen distance and the cantilever deflection and piezo displacement; a cycle of probe oscillation is recorded [68].

### 6.3. Specimen Preparation

The first requirement for proper visualization is the size of the specimen [66]; large specimens cannot be inserted into the scanner. The scanning area is restricted by specification of the microscope. The scan is limited to an area of 240 μm × 240 μm [64]. Moreover, the scanned surface has to be relatively flat, especially when tapping mode is used, because the probe is adapted to limited height fluctuations (usually below 20 μm). Both parameters (the scan area and height fluctuations) are closely related to the parameters of piezoelectric scanner, being the main part of all scanning probe microscopes used for surface analysis. The aspect of surface smoothness remains controversial. According to several studies, grinding and polishing enamel before the observation serves as a guarantee of specimen standardization [64,69]. An opposing recommendation is to use untreated enamel in dental research when it is possible, because the pretreatment may lead to artefacts in the uppermost areas of the specimen [70]. The external enamel layer is then less mineralized and scratches are more visible [70].

### 6.4. AFM in Dental Adhesion Research

The search in PubMed MEDLINE revealed almost 870 papers including words “AFM” OR “atomic force microscopy” AND “dental” from the last decade. Because the etching protocol is crucial for effective bonding, it became a common aim in AFM studies. The nanoscale measurements provided by AFM were necessary to reveal the dependence of adhesion strength on surface roughness. The roughness was measured according to the increased time of acid etching [71]. Research confirmed that the etching pattern is responsible for micromechanical retention, but increased etching time does not improve the adhesive properties [71,72]. This conclusion is in agreement with previous SEM investigations conducted by Wang et al. who showed no significant difference in microtensile bond strength when enamel specimens were etched for 15 or 30 s [73]. In nanoscale images obtained through AFM, it was clearly visible that prolonged etching generated changes in external layers of enamel without any improvement in adhesion and the quality of enamel was impaired with decreased tissue thickness. The longer the etching time, the greater the loss of dental tissue, without an increase in resin retention [71].

As a promising alternative to acid etching, some researchers considered laser irradiation. Indeed, an AFM study was conducted to assess the enamel roughness after exposition to Er:Yag radiation (2970 nm wavelength, 1.2 W) [74]. The images showed that the surface roughness is comparable to conventional etching. The effect provided sufficient microretention for adhesive resins without undesirable superficial demineralization characteristics for acid action. There were also differences in surface morphology—acid etching created regular, visible grooves, whereas after laser irradiation the layer was uneven, non-homogenous and characterized by microcracks. Both patterns were favorable for adhesion (these results were confirmed by shear bond strength test), but the surface after acid etching is more susceptible to caries due to demineralization [74]. The nanostructure of resin itself can also exert a considerable impact on bond strength. It was confirmed with AFM that higher roughness of resin surface creates larger contact area with enamel and larger numbers of resin tags can penetrate microcavities formed under the action of acids [75].

Thanks to a precise visualization of surface topography, AFM was used to evaluate the Young’s modulus of elasticity (Ei) on the dentin–resin interface after application of remineralization-inducing agents. The beneficial effect of primers with biomimetic analogs combined with ion-releasing resin was assessed with three modern microscopic techniques—CLSM, SEM and AFM. Moreover, the observation was supported with spectroscopic methods. AFM revealed a significant increase of Ei in the middle and bottom parts of the hybrid layer after application of biomimetic analogs. Dye-assisted CLSM imaging confirmed the presence of a sound dentin–resin interface. SEM was used after resin debonding and revealed the residual collagen fibrils and fractured resin tags in dentin tubules. Additionally, the results obtained from spectroscopy confirmed the presence of areas of remineralization. These findings demonstrated the promising remineralizing effect of biomimetic analogs with the perspective of implementation in restorative dentistry [10].

AFM was also applied in research evaluating the post-treatment condition of enamel with regard to temporary bonded attachments [76,77]. Non-permanent dental elements, such as orthodontic brackets, have to be safely removed (de-bonded) at the end of treatment. Several debonding and cleaning-up protocols were tested in the context of re-establishing the enamel smoothness [77,78]. In such research, the advantage of AFM is the quantification and the possibility of re-observation of the specimen. In such dental research the microscope worked in the contact mode and revealed the surface topography. The data were collected while the tip was moving across surface providing the measurements of average roughness, root mean square roughness and maximum roughness depth [71].

### 6.5. Limitations

The main disadvantage of conventional AFM is the restricted scanning area that can be analyzed; attaining sufficient temporal resolution of larger areas is time-consuming. The need to overcome these limitations resulted in the development of high-speed AFM and large-scan AFM techniques. However, the interaction between the tip and specimen can lead to image artifacts because of the elastic deformation of the specimen’s surface, so the important aspect is the proper adjustment of imaging parameters, scanning mode and type of a probe [3,7,66,69,75].

### 6.6. Future Perspectives

The invention and development of AFM provides a great impetus for the progress of material science. Among numerous implementations, such a technique provides multiparametric description of the polymers that are omnipresent in dentistry. During the last decade new imaging modes (high-speed scanning, infrared spectroscopy, multifrequency imaging) extended AFM functionality [79]. A promising method in adhesion research is single molecule adhesive force microscopy (SMFS), providing the characteristics of strength and conformation of bonds in the molecule as well as intermolecular interactions [80]. AFM-based nanomechanical mapping modes (AFM-NMM), including force volume and modulation, friction force microscopy, contact resonance or nanoindentation, evaluate properties that are crucial in dental material applicability [79]. AFM combined with infrared spectroscopy enables the chemical identification at nanoscale and opens up possibilities for new generation of dental restorations.

## 7. Conclusions

CLSM, SEM, TEM and AFM can be used for several purposes and in several ways in dental research. Among numerous applications of microscopic techniques, particularly valuable are studies concerning dental adhesion (Table 5).

Adhesive materials are present in most areas of dentistry, especially in the restorative, prosthetic and orthodontic treatment. Microscopic observations disclose an interphase relationship between dental hard tissues and bonded material. It helps to improve the quality of the adhesive, which is crucial for orthodontic brackets, composite restorations, veneers, onlays and prosthetic crowns. In micro- and nano-scale, provided by SEM and AFM, the most favorable etching pattern can be observed. Factors which are responsible for potential destruction of enamel or material degeneration can be recognized and avoided in clinical practice.

During the last twenty years there has been a constant growth of the implementation of microscopic techniques in dental research. SEM microscopy combined with another methods (e.g., spectroscopy) is the most popular technique and the number of new publications relevant to the dental area exceed a few hundred each year (Figure 13).

The way to overcome the limitations of each single microscopic method is the combination of techniques so as to gain as much possible information from the observed specimens. The examples of this multi-method dental research are presented in Table 6.

Of all the mentioned techniques, the greatest potential can be seen in atomic force microscopy because of the highest possible magnitude and image resolution. Although AFM is still not as popular a method as SEM, AFM analysis provides comprehensive enamel characteristics including surface roughness and microdamage at nano-level. It is worth emphasizing that there are no specific requirements for specimen preparation. Observation in AFM can be performed without specimen dehydration. It reduces the risk of potential artefacts connected with fixation and drying. CLSM and TEM remain undoubtedly useful in some particular implementations. In research focused on the interconnection between two phases, the use of fluorescent dye in the CLSM technique can be a promising method, especially when extremely high magnification is not required. Precise surface topography can be obtained with TEM, which is still more popular and more easily accessible than AFM.

There is a wide potential for further implementation of microscopic techniques in dental research. Along with spectroscopic methods it will be possible to obtain reproducible quantitative and qualitative analysis of tissue components. This may represent an important source of information about possible iatrogenic enamel damage during adhesive procedures. Observations can provide clinically relevant conclusions and recommendations which can be easily introduced as an enamel-safe bonding and debonding protocol. All described microscopic techniques also give an insight into the remineralization process, incorporating the latest trends in application of bioactive materials and their interaction with dental tissues.

Each microscopic technique implemented as a single method has its own limitations, so the multi-method approach should serve as a research standard. The overall protocol in adhesion research can be recommended as follows: CLSM as a non-invasive method that should be used in some specific projects as the first step of investigation. It can provide the general overview of surface topography as well as for visualization of rapid chemical and biochemical features on the specimens. For instance, in dentistry, a researcher can obtain valuable information about the depth of the penetration of etching agents and adhesive systems within dental tissues. It is not an overstatement to affirm that electron microscopy may serve as a gold standard in microscopy for dental research. Indeed, SEM provides a thoughtful analysis of the surface’s morphology. Any defect at the tissue-restoration interface can be disclosed in high-resolution image. TEM ideally complements the SEM observations, providing the view of the internal structure of the specimen. Particularly important is the visualization of dentinal collagen and the structure of the hybrid layer—elements principally involved in the adhesion to dentin. However, due to the fact that specimens cannot be re-used after preparation for electron microscopy observation, AFM may be implemented as a valuable alternative to SEM and TEM, especially when it is necessary to assess the surface roughness and nanomechanical properties in natural conditions (i.e., immersed in water or body fluids).

## Figures and Tables

**Figure 1 biosensors-11-00408-f001:**
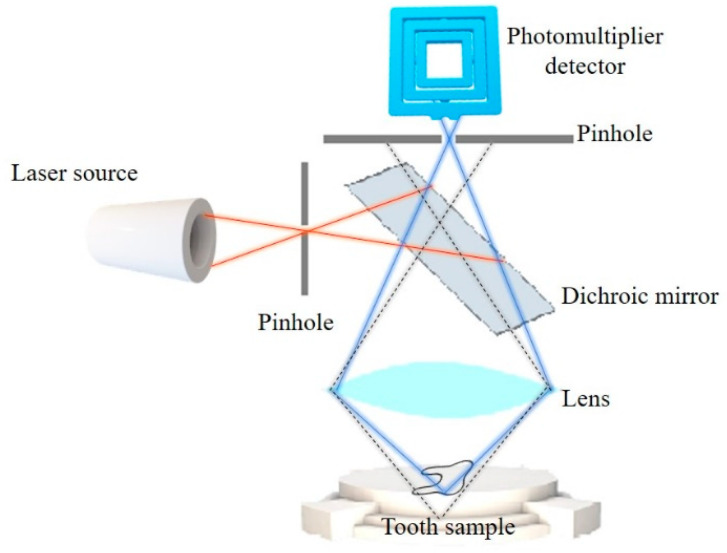
Principle of CLSM.

**Figure 2 biosensors-11-00408-f002:**
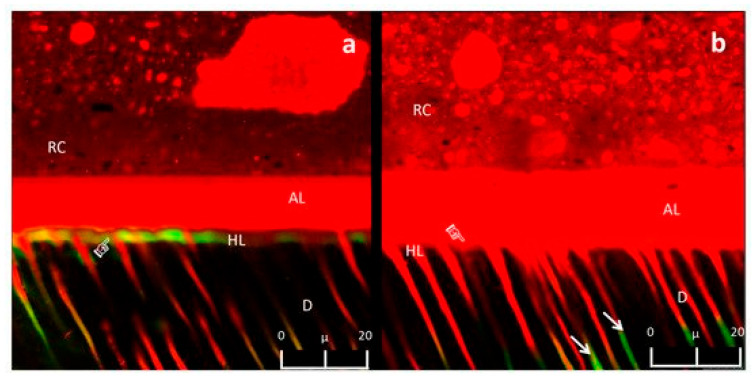
CLSM images of dentin (D) (63×, zoom 2×, depth: 10 µm) primarily exposed to the pretreatment solutions followed by an application of bonding system (AL) and resin composite (RC). The presence of fluorescein was observed in hybrid layer (HL) (**a**) and in the dentin tubules (**b**) depending on the tested group of flavonoids. From: Dávila-Sánchez, A.; Gutierrez, M.F.; Bermudez, J.P.; Méndez-Bauer, L.; Pulido, C.; Kiratzc, F.; ... and Arrais, C.A.G. (2021). Effects of Dentin Pretreatment Solutions Containing Flavonoids on the Resin Polymer-Dentin Interface Created Using a Modern Universal Adhesive. Polymers, 13(7), 1145. [14].

**Figure 3 biosensors-11-00408-f003:**
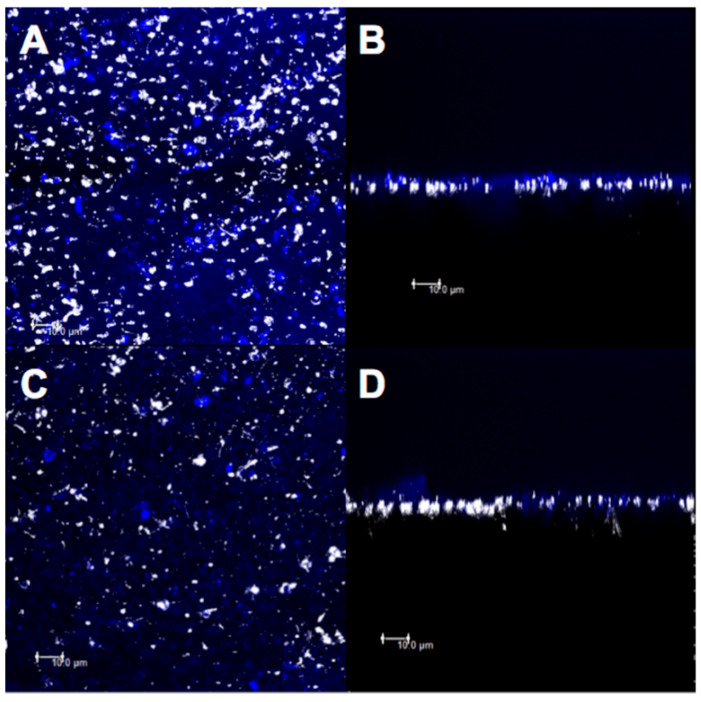
Confocal laser scanning microscopy images (fluorescence mode) in XY and XZ axis of dentin surface. (**A**,**B**) Fluoride varnish with nanofiber and (**C**,**D**) after 6% citric acid challenge. From: Bastos-Bitencourt, N.; Velo, M.; Nascimento, T.; Scotti, C.; Fonseca, M.G.D.; Goulart, L.; ... and Sauro, S. (2021). In Vitro Evaluation of Desensitizing Agents Containing Bioactive Scaffolds of Nanofibers on Dentin Remineralization. Materials, 14(5), 1056. [16].

**Figure 4 biosensors-11-00408-f004:**
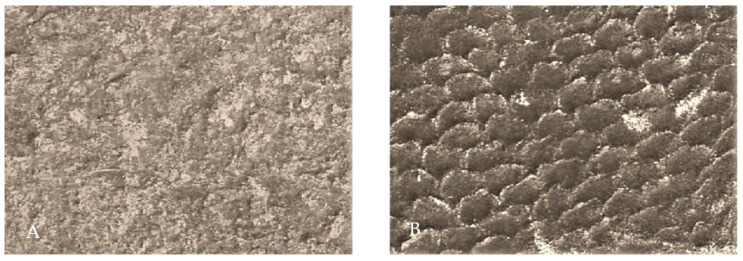
CLSM images of enamel: (**A**) unetched; (**B**) etched for 30 s H_3_PO_4_ (courtesy of S. Sauro).

**Figure 5 biosensors-11-00408-f005:**
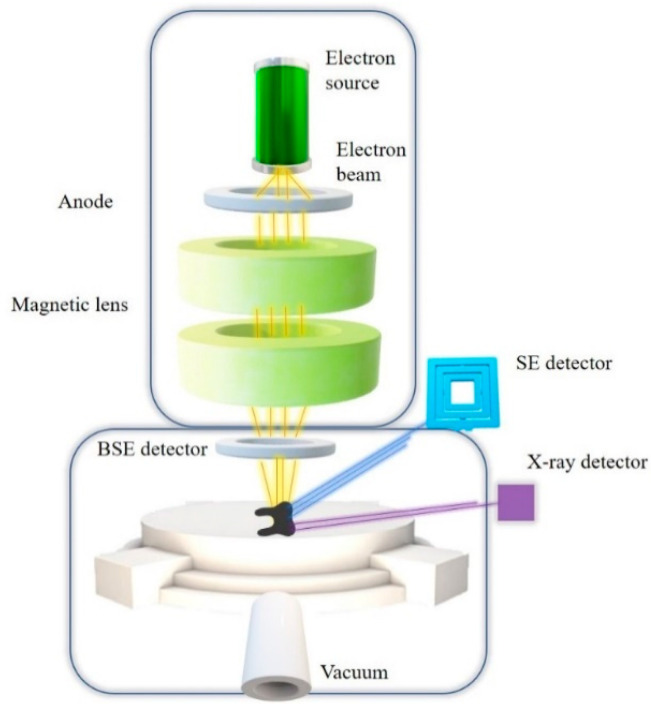
Principle of SEM.

**Figure 6 biosensors-11-00408-f006:**
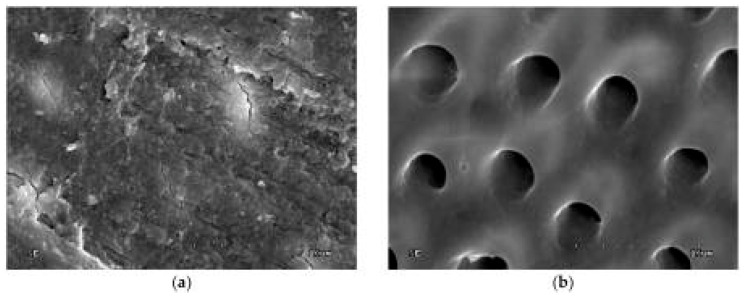
Scanning electron microscopy (SEM) image of dentin surface: (**a**) Prepared with bur, unetched (1000×); (**b**) etched with 36% orthophosphoric acid (5000×). From: Lapinska, B.; Klimek, L.; Sokolowski, J.; and Lukomska-Szymanska, M. (2018). Dentin surface morphology after chlorhexidine application—SEM study. Polymers, 10(8), 905 [27].

**Figure 7 biosensors-11-00408-f007:**
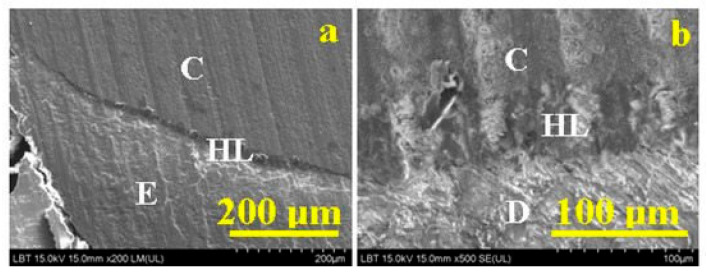
SEM images for (**a**) composite (C)—enamel interface (E); (**b**) composite (C)—dentin (D) interface; HL—hybrid layer. From: Chisnoiu, A.M.; Moldovan, M.; Sarosi, C.; Chisnoiu, R.M.; Rotaru, D.I.; Delean, A.G.; ... and Pastrav, M. (2021). Adaptation Assessment for Two Composite Layering Techniques Using Dye Penetration, AFM, SEM and FTIR: An In-Vitro Comparative Study. Applied Sciences, 11(12), 5657 [28].

**Figure 8 biosensors-11-00408-f008:**
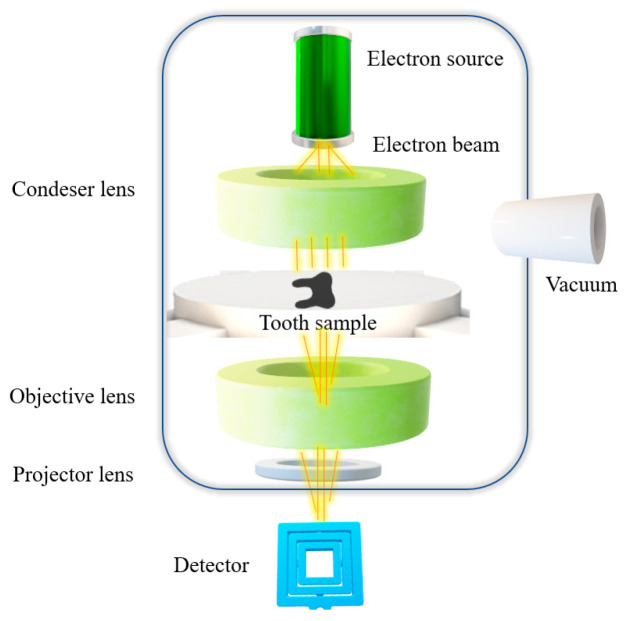
Principle of TEM.

**Figure 9 biosensors-11-00408-f009:**
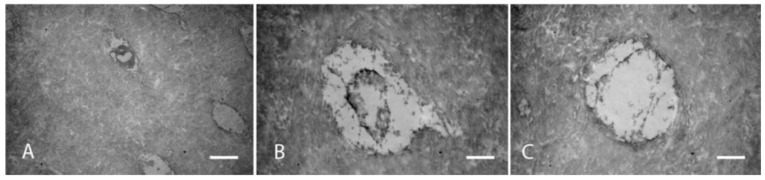
TEM analysis of dentin (**A**–**C**): Compacted collagen with a fibrillar scaffold and regular odontoblastic process in healthy donors. From: Josic, U.; Maravic, T.; Bossù, M.; Cadenaro, M.; Comba, A.; Ierardo, G.; ... and Mazzoni, A. (2020). Morphological Characterization of Deciduous Enamel and Dentin in Patients Affected by Osteogenesis Imperfecta. Applied Sciences, 10(21), 7835 [49].

**Figure 10 biosensors-11-00408-f010:**
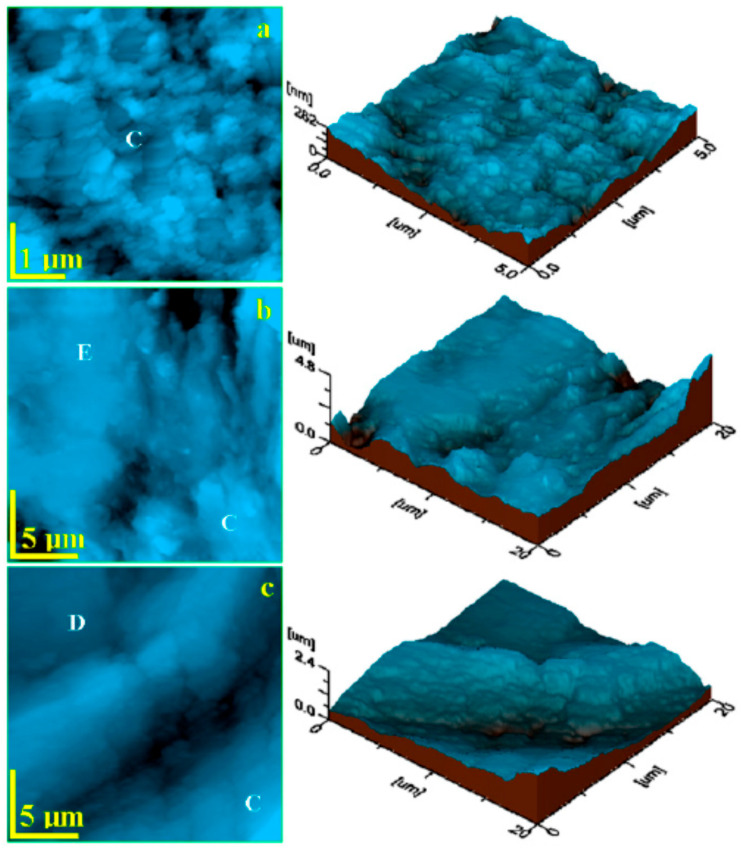
AFM 2D topographic of: (**a**) composite (restorative material), (**b**) composite–enamel interface, and (**c**) composite–dentin interface. Tri-dimensional images are given next to 2D topographic images. C—composite, E—enamel, D—dentin. From: Chisnoiu, A.M.; Moldovan, M.; Sarosi, C.; Chisnoiu, R.M.; Rotaru, D.I.; Delean, A.G.; ... and Pastrav, M. (2021). Marginal Adaptation Assessment for Two Composite Layering Techniques Using Dye Penetration, AFM, SEM and FTIR: An In-Vitro Comparative Study. Applied Sciences, 11(12), 5657 [28].

**Figure 11 biosensors-11-00408-f011:**
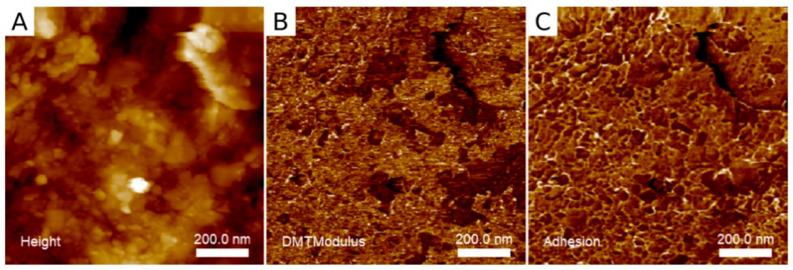
AFM mapping of dental enamel after orthodontic treatment ((**A**)—topography; (**B**)—mean Young’s modulus and (**C**)—adhesion). From: Machoy, M.; Wilczyński, S.; Szyszka-Sommerfeld, L.; Woźniak, K.; Deda, A.; and Kulesza, S. (2021). Mapping of Nanomechanical Properties of Enamel Surfaces Due to Orthodontic Treatment by AFM Method. Applied Sciences, 11(9), 3918 [65].

**Figure 12 biosensors-11-00408-f012:**
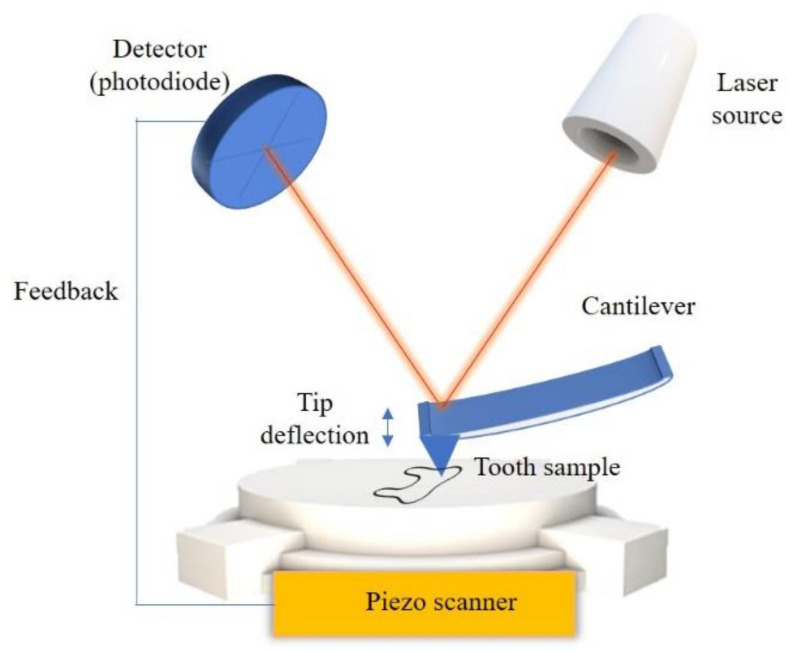
Principle of AFM.

**Figure 13 biosensors-11-00408-f013:**
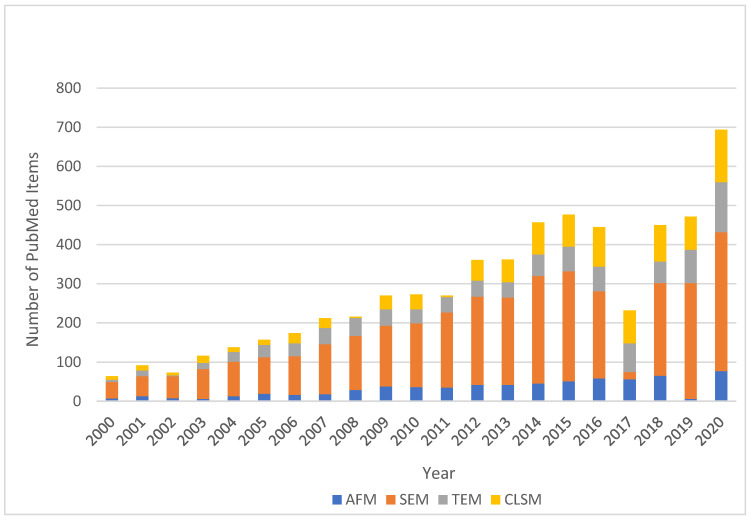
Histogram of the articles published in the years 2000–2020 according to PubMed on the basis of search phrases “AFM”, “SEM”, “TEM” and “CLSM” connected with the phrase “dental”.

**Table 1 biosensors-11-00408-t001:** Comparison of traditional microscopic techniques [1,4,5,6,7].

Technique	Optical Microscopy			Electron Microscopy		Scanning Probe Microscopy
	Conventional	Fluorescent	Confocal Scanning Microscope	Transmission Electron Microscope	Scanning Electron Microscope	Atomic Force Microscope
Source of energy	light (380–750 nm)	Specific light wavelength (absorbed by the fluorophores in the specimen)	Localized laser excitation and fluorescence effect	The interaction of beam of electrons that is transmitted through the specimen	Beam of high-energy electrons that interact with the specimen surface and generate a variety of signals	Deflection of the tip caused by atomic and electrostatic forces between the tip and specimen
Magnification	To 100–1000×			2,000,000–50,000,000×		To 100,000,000×
Resolution	100–1000 nm			1–10 nm		0.2 nm (horizontal) 0.05 nm (vertical)
Main implementations	adhesive interface integrity the presence and characteristic of dentinal smear layer the depth of penetration of various materials in dentinal tubules invasiveness of etchants			surface topography and adhesive interface presence of microcracks in the tissue surface and interfacial gaps mechanical wear of dental materials and tissues adhesive and cohesive failures internal nanostructure (TEM)		surface roughness and its impact on dental adhesion preferable in assessment of dentinal structure (undestroyed dentinal collagen network

**Table 2 biosensors-11-00408-t002:** Search strategy.

Search Strategy	Inclusion Criteria	Exclusion Criteria
Type of paper	Systematic reviews, metanalyses, original research papers published	-
Form of publication	Full-text or book chapter	Abstract only
Language of publication	English	Non-English publications
Publication content	Relevant to the subject of dentistry and dental research	Not including the content associated with dental tissues or dental materials

**Table 3 biosensors-11-00408-t003:** Protocol before microscopic observation [2,3,8,10,11].

Procedure	CLSM	SEM, TEM	AFM
**Sectioning**	Water-cooled diamond wafering blade		
**Cleaning and polishing**	The removal of external enamel layer with a water-cooled rotating polishing machine (automatic metallurgical grinder and polisher) Sequence of silicon carbide polishing discs (200 rpm) ○ 10 s of 600-grit ○ 20 s of 1200 grit ○ 30 s of 2500 grit ○ 60 s of 4000 grit 1 min of ultrasonic water bath after use of silicone carbide polishing disc		
**Specific recommendations according to the method**	Labelling with fluorescent dye (i.e., rhodamine B)	Dehydratation and fixation Covering with electro-conducting coating layer	Flat specimen surface is necessary (Specimen labelling, dehydration, fixation and covering is not required)
**Specimen mounting**	To the glass slide with cyanoacrylate glue	To the aluminum stub with double-sided carbon or copper tape	With a double-sided tape/glue to the steel disk that is finally mounted into magnetic specimen

**Table 4 biosensors-11-00408-t004:** Application of SEM in dental research in year 2020.

Area of Dentistry	Number of Items in PubMed Database (Year 2020)	Subject of SEM Research
Restorative dentistry, prosthetics and dental materials	206	microstructural characteristics of various dental materials–glass ionomer cements, resin composites with different fillers, cad-cam restorations, zirconia-based ceramics (163 items) surface preparation protocols, etching methods and its effect on adhesive bonding (22 items) assessment of enamel/restoration surface after various polishing protocols (5 items) biofilm formation on various restorative materials (7 items) marginal adaptation (5 items) enamel remineralization (3 items)
Biomaterials and guided tissue regeneration	26	scaffolds for potential application in bone engineering (26 items)
Implantology	56	biofilm on implant surface (10 items) mechanical resistance of dental implants connection between dental implant and abutment (30 items) implant surface in context of osteointegration (16 items)
Orthodontics	12	bond strength of metallic/ceramic orthodontic brackets to enamel, acrylic, or porcelain surfaces (2 items) coating on orthodontic arch-wires (3 items) corrosion of orthodontic arch-wires (1 item) the force degradation and deformation of the open-closed and open springs of NiTi (1 items) enamel condition after orthodontic debonding (5 items)

**Table 5 biosensors-11-00408-t005:** Advantages and disadvantages of CLSM, SEM and AFM in dental research.

	CLSM [4,11,12,17,78]	SEM [17,30,69,78]	TEM [39,41,42]	AFM [3,7,66,69,75]
Advantages	out-of-focus is signal reduced good contrast due to fluorescence optical sectioning of specimen with 3D reconstruction	high quality image of surface topography and adhesive interface higher resolution and 104× higher magnification cf. CLSM visualization of larger area of specimen (few millimeters) shorter time to obtain an image cf. AFM can be easily combined with spectroscopic methods and profilometry	high quality image of internal structure of the specimen higher resolution and 104× higher magnification cf. CLSM	highest possible magnification and image resolution does not require vacuum environment and methods of specimen coating. fixation, dehydration does not require fluorescent dye reobservation of specimen possible
Disadvantages	low magnification (max. 1000×) resolution limited to the optical diffraction limit	non-conducting specimens (i.e., dental tissues) require coating dehydration and fixation of biological specimens may cause artifacts lower contrast cf. CLSM	non-conducting specimens (i.e., dental tissues) require coating dehydration and fixation of biological specimens may cause artifacts ultrathin sections of specimen are required	few minutes require to obtain the image of dental tissues strong restriction to flatness of a specimen (to provide the proper access of tip) restricted scanning area

**Table 6 biosensors-11-00408-t006:** The examples of multi-method microscopic research in dentistry [16,81,82,83,84].

Author, Year	Research Topic	Microscopic Techniques	Observed Parameters	Authors’ Conclusions
Wu, et al. (2020) [81]	Remineralization of caries lesions in dentin after application of bioactive glass (BAG)	AFM, CLSM	AFM—surface topography, microhardness, the depth of the remineralization CLSM—fluorescence on the superficial layer of the lesion (decrease in fluorescence correlated with remineralization)	BAG obtains a promising remineralization effect
Olley, et al. (2020) [82]	Dentin tubule patency and surface roughness after novel dab-on or brushing abrasion	CLSM, AFM, SEM, EDX, contact profilometry (CP)	CP—dentinal surface roughness CLSM—comparison of tubular patiency, AFM-roughness in intertubular region, SEM –the penetration of dentifrice EDX—constituents of deposits existing in dentinal tubules	Dab-on applications of either SnF2 or NaF dentifrice reduce the patency of dentine tubules and therefore reduce dentinal hypersensitivity
Šugár et al. (2020) [83]	Laser machining of Ti-graphite composite for dental application	CLSM, SEM, EDX	CLSM, SEM—visualization of laser-prepared surface, EDX—the elemental composition	The thermal energy from laser put to the Ti-graphite composite has a positive effect on surface properties of dental implant
Bastos-Bitencourt, et al. (2021) [16]	The effect of bioactive scaffolds of nanofibers on dentin remineralization	CLSM, SEM	CLSM—identification and comparison the dentinal tubule obliteration (magnification 40×) SEM—revealing the dentin morphology (magnification 3000×)	Desensitizing agents with nanofibers are potentially effective in dentin remineralization
Pandele, et al. (2020) [84]	The method of synthesis new composite films based on polylactic acid and micro-structured hydroxyapatite particles (HA)	SEM (AFM) microscopy, FT-IR and, Raman spectroscopy, thermogravimetry (TG)	SEM, AFM—morphological analysis FT-IR and Raman spectroscopy- structural analysis TG—assessment of thermostability of the polymer.	The crystallinity of the composite films was decreased in comparison to the pure polymer. The presence of hydroxyapatite crystals did not have a significant influence on the degradation temperature of the composite film.

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
