# Peer review of "Traditional Microscopic Techniques Employed in Dental Adhesion Research—Applications and Protocols of Specimen Preparation"

_biosensors, 2021, doi:10.3390/bios11110408_

Round 1

Reviewer 1 Report

Please check my comments in the file attached

Regards

Author Response

Dear Sir or Madam,

Thank you for your review. We are very thankful for all your comments that enhanced the quality of manuscript. All your suggestions were addressed accordingly: the line is given (in Track Changes - Final: Show Markup).

Comment 1:

Line 63: I suggest to change the headline of table 1 and use the word “traditional” instead of “contemporary” –according to the main title of the whole manuscript, because, for instance, “optical microscopy” is not a modern technique.

- However, the introduction (chapter 1) is slight “poor in content”. Please try to improve it.

Answer: Thank you for your comment. The headline of the table 1 was revised by changing the word “contemporary” into “traditional”  (line 83) The change in manuscript was done as well (line 54, Introduction section). Moreover, to improve the contented introduction, the additional paragraph was added to so as to explain the significance of the of our paper for researchers who are planning to apply microscopic techniques in their studies. Lines 58-71)

Comment 2  

 In confocal microscopy, please emphasize the fact that such a technique does not require special preparation of specimens so they are observed with no destructive process

- Conversely in electron microscopy, please emphasize the fact that such techniques require “destructive” special preparation of specimens so they cannot be reused anymore.

Answer: Thank you for your comment. The paper was revised to emphasize the aspects highlighted by reviewer. The change in manuscript was done as follows:

  1. According to each technique additional chapters entitled “Limitations” were added (line 252 :CLSM; lines 448- limitations of SEM, line 568 - limitations of TEM, line 725 limitations of AFM)
  2. The specimen preparation is summarized in the table 3. The text remains unchanged to fulfill the important aim of the paper – presentation of the specimen preparation protocols
  3. The advantages and disadvantages of each technique are summarized in table 5
  4. The problem of possible specimen destruction is also emphasized in conclusion section (lines 801-815)

Comment 3:  For TEM………please add a chapter dedicated to the special staining technique required especially for collagen and/or remineralisation analysis

Answer: Thank you for your comment. The description of TEM technique was enriched with the description of negative staining method (lines 501-511 and 551-554) that can be the dedicated to dentin collagen.

Comment 4: Line 101 – I suggest adding one sentence explanation of term “optical sectioning”

Answer: Thank you for your comment. The term optical sectioning is explained in lines 114-117

Comment 5:

Please include more explanation according the possible implementation AFM technique in adhesion mapping , calculating Young’s modulus and friction measurements (with the reference to the figure 8)

Answer: Thank you for your comment. In AFM technique we paid particular attention to force volume mapping (lines 650-658).  

Comment 5: I feel that with this complete research the authors may propose some recommendations and question if the use of all the techniques together is not the excellence for investigations with dental adhesives. Or, maybe, at least, the combination of 3 of them. Again, in the conclusion section, I felt a lack of opinion provide by the authors.

Answer: Thank you for your comment. The recommendations for using combined microscopic techniques in dental adhesion studies are now included in the final paragraph of Conclusion section (lines 798-811)

All lines required font editing were corrected according to your suggestions.

We are beholden for the time and effort that you have devoted to give a valuable response to our submission. We are grateful for insightful remarks on our paper.

Reviewer 2 Report

Dear authors: 
The manuscript entitled "Traditional microscopic techniques employed in dental adhesion research – applications and recommendations for specimen preparation." shows a complete summary of the use of traditional microscopy equipment such as confocal microscopy, SEM, TEM and AFM in applications with dental materials for adhesion. The description and a brief historical summary were well-design. However, the manuscript lacks innovative content or new information from the all search performed. The manuscript at this moment seems like a book chapter in my view. The authors need to find new knowledge from all this systematic research and show it to the readers. 

Some recommendations for the next submission:

1. The recommendations mentioned in the title of this manuscript about the preparation of samples were not clear. The manuscript showed only protocols described by other authors. I didn't see a final conclusion provided by the authors of what is better or what is worst. Or, the title needs to change.

2. Each chapter needs an in-deep analysis of limitations, advantages, examples of applications and future perspectives of each technique (SEM,TEM,AFM,Confocal). More than a description of historical background and samples preparation. In my view, the samples preparation can be summarized in one large table and the text about this topic can be reduced. Leaving text only for the author's conclusions about each technique.

3. The lack of scientific new content, maybe, can be fulfilled with one table showing the select articles and which techniques were applied (one or more). Also, with what aim the techniques were applied and the conclusions achieved by each author.

4. The example figures of each technique are so simple, only one article was used for each technique. The authors can show more examples of the use of each technique in one same figure, using parts A,B,C,D.

5. I feel that with this complete research the authors may propose some recommendations and question if the use of all the techniques together is not the excellence for investigations with dental adhesives. Or, maybe, at least, the combination of 3 of them. Again, in the conclusion section, I felt a lack of opinion provide by the authors.

With these points, I can't recommend the manuscript to a high-quality Journal like Biosensors.

Author Response

Dear Sir or Madam,

Thank you for your review. We are very thankful for all your comments that enhanced the quality of manuscript. All your suggestions were addressed accordingly: the line is given (in Track Changes - Final: Show Markup).

Comment 1: The recommendations mentioned in the title of this manuscript about the preparation of samples were not clear. The manuscript showed only protocols described by other authors. I didn't see a final conclusion provided by the authors of what is better or what is worst. Or, the title needs to change.

Answer: Thank you for your comment. The title was revised by changing the word “recommendations” into “protocols” The change in manuscript was done as well (line 79, Introduction section). Moreover, the additional paragraph was added to so as to explain the significance of the of our paper for researchers who are planning to apply microscopic techniques in their studies (Lines 58-71). The conclusions of this study were also revised. However, we invite this reviewer to consider that this is a narrative review paper and not an experimental research report one, hence we can only summarize the main concepts derived from all those articles that we have analyzed and included in the references list  

Comment 2  Each chapter needs an in-deep analysis of limitations, advantages, examples of applications and future perspectives of each technique (SEM, TEM, AFM, Confocal). More than a description of historical background and samples preparation. In my view, the samples preparation can be summarized in one large table and the text about this topic can be reduced. Leaving text only for the author's conclusions about each technique.

Answer: The paper was revised to emphasize the aspects highlighted by this reviewer. The change in manuscript was performed as follows:

  1. According to each technique two additional chapters were added – “Limitations” and “Future perspectives” (lines 252-289: limitations and future perspectives of CLSM; lines 445-451: limitations of SEM; 567-576: limitations of TEM, lines 577-794: future perspectives of SEM and TEM summarized together, 725-746: limitations and future perspectives of AFM)
  2. The specimen preparation is summarized in the table 3. The text remains unchanged to fulfill the important aim of the paper – presentation of the specimen preparation protocols
  3. The advantages and disadvantages of each technique are summarized in table 5
  4. The examples of application are included in the separate chapters ( “… in dental adhesion research”) The text was enriched with more exemplar applications (lines 190-194, lines 390-394, 613-619) and table 6 in Conclusions section (line 776)

Comment 3:  The lack of scientific new content, maybe, can be fulfilled with one table showing the select articles and which techniques were applied (one or more). Also, with what aim the techniques were applied and the conclusions achieved by each author.

Answer: Thank you for your comment. Recommended table was added in Conclusion section (table 6, line 775) The table includes five selected papers that were published in 2020-2021 showing the research combining microscopic techniques, the aim of multi-method approach and author’s conclusions. Moreover, the description of TEM technique was enriched with the description of negative staining method (lines 501-511 and 551-554) that can be the dedicated to dentin collagen. In AFM technique we paid particular attention to force volume mapping (lines 650-658). The total number of references extended from 65 to 85.

Comment 4. The example figures of each technique are so simple, only one article was used for each technique. The authors can show more examples of the use of each technique in one same figure, using parts A,B,C,D.

Answer: Thank you for your comment. According to the suggestions, the described techniques are illustrated with the precisely selected figures. Illustrations are reproduced from more than one article to provide a better view about each technique. We decided to attach each figure separately, because if they were presented as a one figure,  the image quality would be insufficient. (figure 3 – lines 195-201; figure 6 – lines 389-394, figure 9 – lines 612-619)

Comment 5: I feel that with this complete research the authors may propose some recommendations and question if the use of all the techniques together is not the excellence for investigations with dental adhesives. Or, maybe, at least, the combination of 3 of them. Again, in the conclusion section, I felt a lack of opinion provide by the authors.

Answer: Thank you for your comment. The recommendations for using combined microscopic techniques in dental adhesion studies are now included in the final paragraph of Conclusion section (lines 798-811)

We are grateful for insightful remarks on our paper and for the time and effort that this reviewer has spent to give a valuable response to our submission. We hope that the changes we have made can be satisfying, so making this paper acceptable for publication.

Many thanks and best wishes.

Round 2

Reviewer 2 Report

Dear authors: 

The improvements really addressed all recommendations. For a narrative review, now you have excellent information and pictures of the traditional microscopic techniques.

Table 6 really show important scientific information showing the application in complex studies. Moreover, the final conclusions showed an interesting view of the author's knowledge.

Kind regards.